# Solar Radiation Nowcasting Using a Markov Chain Multi-Model Approach

Xinyuan Hou [1,2,*], Kyriakoula Papachristopoulou [1,3,4], Yves-Marie Saint-Drenan [5] and Stelios Kazadzis [1]

[1] Physikalisch-Meteorologisches Observatorium Davos/World Radiation Center (PMOD/WRC), 7260 Davos Dorf, Switzerland; kpapachr@noa.gr (K.P.); stelios.kazadzis@pmodwrc.ch (S.K.)
[2] Department of Physics, ETH Zürich, 8093 Zürich, Switzerland
[3] Laboratory of Climatology and Atmospheric Environment, Sector of Geography and Climatology, Department of Geology and Environment, National and Kapodistrian University of Athens (LACAE/NKUA), 15772 Athens, Greece
[4] Institute for Astronomy, Astrophysics, Space Applications and Remote Sensing, National Observatory of Athens (IAASARS/NOA), 11810 Athens, Greece
[5] O.I.E. Centre Observation, Impacts, Energy, MINES ParisTech, PSL Research University, 06904 Sophia Antipolis, France; yves-marie.saint-drenan@mines-paristech.fr
[*] Correspondence: xinyuan.hou@pmodwrc.ch

**Abstract:** Solar energy has found increasing applications in recent years, and the demand will continue to grow as society redirects to a more renewable development path. However, the required high-frequency solar irradiance data are not yet readily available everywhere. There have been endeavors to improve its forecasting in order to facilitate grid integration, such as with photovoltaic power planning. The objective of this study is to develop a hybrid approach to improve the accuracy of solar nowcasting with a lead time of up to one hour. The proposed method utilizes irradiance data from the Copernicus Atmospheric Monitoring Service for four European cities with various cloud conditions. The approach effectively improves the prediction accuracy in all four cities. In the prediction of global horizontal irradiance for Berlin, the reduction in the mean daily error amounts to 2.5 Wh m$^{-2}$ over the period of a month, and the relative monthly improvement reaches nearly 5% compared with the traditional persistence method. Accuracy improvements can also be observed in the other three cities. Furthermore, since the required model inputs of the proposed approach are solar radiation data, which can be conveniently obtained from CAMS, this approach possesses the potential for upscaling at a regional level in response to the needs of the pan-EU energy transition.

**Keywords:** solar radiation nowcasting; solar energy prediction; Markov chain models

## 1. Introduction

Global horizontal irradiance (GHI) is defined as the downwelling solar irradiance observed at ground level on horizontal surfaces and integrated over the whole shortwave spectrum. It is the sum of the direct and diffuse irradiances that come from the direction of the sun and the rest of the sky, respectively [1]. Global solar irradiation data are crucial for renewable energy applications. including the planning, monitoring, and forecasting of photovoltaic (PV) systems. However, irradiation data are not readily available everywhere, especially when meteorological stations are lacking in isolated areas [2]. Ngoko et al. [3] also noted that historical solar radiation data taken at high sampling frequencies are unavailable for many locations, thereby limiting some just-in-time applications for specific sites where such data may not exist.

As with the majority of meteorological variables, a wide spectrum of forecasting techniques find current application in solar radiation forecasting. One group of solar forecasting methods consists of statistical solar forecasts based on irradiance time series [4] or machine learning methods such as artificial neural networks (ANN) [5]. Another group

includes physical methods, such as irradiance forecasting, using either cloud motion vectors (CMV) [6] or numerical weather prediction. Both groups also extensively employ post-processing methods, such as the use of model output statistics [5], quantile forecasts [7], and ensemble prediction systems [8]. Statistical post-processing techniques learn error patterns by comparing forecasts and observations in order to reduce the error in the final prediction [9].

Surface solar radiation is highly variable, mainly driven by synoptic and local weather patterns. While clear-sky irradiance is strongly influenced by atmospheric composition, including aerosol and water vapor, all-sky irradiance changes with clouds, resulting in a high fluctuation of solar radiation. Considering the variability of cloudiness and solar irradiance, probabilistic techniques, such as Markov-chain-based methods, are widely used in solar forecasting. For example, a homogeneous recurrent Markov process with discrete states aided in predicting sunshine and cloud cover for Payern, Switzerland and Perth, Australia [10]. Hocaoglu et al. [11] combined the Mycielski algorithm with a Markov chain model to forecast hourly solar radiation from historical records for two regions in Turkey. A Markov chain probability distribution mixture model was employed for the forecast of a clear-sky index in Hawaii and Norrköping [12,13].

In addition to solar radiation forecasting, the synthetic generation of solar radiation time series has also been of interest and can be found in some previous studies: Poggi et al. [14] studied the stochastic properties of generated hourly total solar radiation in Corsica, France using a Markov model following a shifted negative binomial distribution. Mellit et al. [2] used an ANN and Markov transition matrices for the generation of daily global solar radiation data. Ngoko et al. [3] generated high temporal resolution (1-minute) solar radiation data using Markov models, with inputs that included hourly sea level pressure, wind speed, cloud-based height, and cloud cover for two locations in Japan. Bright et al. [15] set up a Markov chain with hourly weather observation data for cloud cover in Leeds, U.K. to generate minutely irradiance time series. Shepero et al. [16] employed hidden Markov models with Gaussian observation distributions to generate clear-sky index time series for two locations with different climate conditions (Hawaii and Norrköping). Urrego-Ortiz et al. [17] proposed a Markov chain model for day-ahead forecasting of hourly GHI in Medellín, Colombia.

Nowcasting systems correspond to the forecast of the short-term evolution of the weather, specifically up to 6 h ahead [11]. It is important to provide information about solar radiation availability in the next few hour(s) to facilitate solar energy management by plant operators and grid operators. In this study, we propose a hybrid approach utilizing several forecast techniques, including but not limited to the persistence ensemble and the Markov chain, in order to increase the accuracy of solar radiation nowcasting. Combining different models in a single entity is expected to leverage the strengths of both methods while reducing their weaknesses.

## 2. Data

We use radiation data provided by the Copernicus Atmospheric Monitoring Service (CAMS) radiation service because of its wide geographical coverage from $-66°$ to $66°$ in both latitude and longitude [18]. The CAMS radiation data are output using the Heliosat-4 method [19] and the McClear fast clear-sky model [1,20]. McClear utilizes aerosol properties and the total column content of water vapor and ozone obtained from the CAMS model. Aside from the solar zenith angle and the ground albedo, six other parameters describing the optical state of the atmosphere are used as inputs to McClear look-up tables to estimate downwelling shortwave radiation in cloudless conditions [20]: total column content of ozone, total column content of water vapor, aerosol optical depth at 550 nm, aerosol mixture, atmospheric profile, and elevation of the ground above mean sea level.

Previous studies have identified that interactions between clouds and radiation contribute to the stochastic variations of GHI (e.g., [17]). Here, this study considers the cloud modification factor (CMF), defined as the ratio of GHI in all-sky conditions to that in

clear-sky conditions (GHI$_{CS}$), keeping the same atmospheric composition but assuming the absence of clouds:

$$\text{CMF} = \frac{\text{GHI}}{\text{GHI}_{CS}} \tag{1}$$

This measure is similar to the clearness index, which is a ratio of the solar radiation recorded on the earth's surface to extraterrestrial solar radiation. If CMF approaches 0, the sky is cloudy, while a CMF value of 1 implies that there is a clear sky without clouds.

The data from CAMS are updated until two days prior to the instantaneous query time. We take the time series with an interval of 15 min, extending from 1 February 2004 to 31 December 2020. Thanks to the wide coverage of CAMS data, we choose four cities in Europe with different cloud conditions, listed in Table 1.

**Table 1.** Four chosen cities with their coordinates, long-term cloud modification factor (CMF, mean ± standard deviation from 2004 to 2019), and mean absolute change of CMF from the previous one time step to the present.

| City | Coordinates | Long-Term CMF | $|\mathbf{\Delta CMF}|$ |
|---|---|---|---|
| Athens | 37.98° N, 23.72° E | 0.787 ± 0.245 | 0.049 |
| Bucharest | 44.44° N, 26.08° E | 0.693 ± 0.278 | 0.054 |
| Berlin | 52.52° N, 13.37° E | 0.656 ± 0.267 | 0.067 |
| Helsinki | 60.19° N, 24.93° E | 0.625 ± 0.267 | 0.063 |

The cities in Table 1 are listed in descending order of long-term average CMF. The last column of Table 1 presents the mean absolute change of CMF from the previous one time step to the present as a measure of cloudiness variability. Athens has comparatively low variability in cloudiness, whereas Berlin has the highest variability among the four cities and thus is the most difficult to predict. Figure 1 shows, as an example, the daily mean insolation (GHI integrated during a day) for each month and the monthly mean CMF for Berlin in 2020.

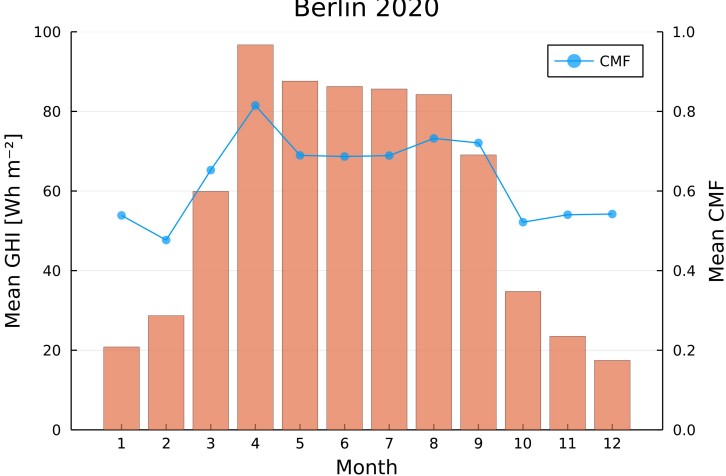

**Figure 1.** Daily mean global horizontal irradiation (GHI) for each month (orange bars, left axis) and monthly mean cloud modification factor (CMF, blue curves, right axis) for Berlin in 2020. The results for the other three cities can be found in Figure A1 in Appendix A.

We divide the CMF time series into three periods: the period from 2004 to 2018 for fitting the transition matrix from historical occurrence (the training set), the year 2019 for evaluating the error metrics of the prediction methods (the validation set), and the year 2020 for testing the performance of different prediction approaches (the test set), including the implementation of the hybrid approach (see Section 3.5).

## 3. Model and Methods

### 3.1. Quantitative Metrics

In this study, we adopt several commonly used metrics for the selection of model parameters as well as accuracy assessment. First, the error of one data pair is defined as the difference between the predicted ($\hat{X}_i$) and the actual value ($X_i$):

$$\epsilon_i = \hat{X}_i - X_i \tag{2}$$

*Standard deviation* is given by:

$$\sigma(\epsilon) = \sqrt{\frac{1}{N} \sum_{i=1}^{N} (\epsilon_i - \bar{\epsilon})^2} \tag{3}$$

where $N$ is the number of data pairs in the time series and $\bar{\epsilon}$ denotes the mean error.

*Pearson Correlation coefficient* is defined as:

$$r = \frac{\sum_{i=1}^{N} (\hat{X}_i - \overline{\hat{X}})(X_i - \overline{X})}{\sigma(\hat{X})\sigma(X)} \tag{4}$$

where $\overline{\hat{X}}$ and $\overline{X}$ represent the mean values of the predicted and the actual time series, respectively.

*Mean bias error* often simply referred to as *bias*, is given by

$$MBE = \frac{1}{N} \sum_{i=1}^{N} \epsilon_i \tag{5}$$

*Mean absolute error* is written:

$$MAE = \frac{1}{N} \sum_{i=1}^{N} |\epsilon_i| \tag{6}$$

*Root mean square error* is defined as:

$$RMSE = \sqrt{\frac{1}{N} \sum_{i=1}^{N} \epsilon_i^2} \tag{7}$$

### 3.2. Markov Chain

A discrete-time Markov chain (MC) describes the transition process of a discrete random variable, where the probability distribution of the variable's following state depends on its previous observations. MC-based methods are frequently used for random variables of finite state space [21]. A first order MC can be described by:

$$P(X_t|X_0, \ldots, X_{t-2}, X_{t-1}) = P(X_t|X_{t-1}) = q_{i_1 i_0} \tag{8}$$

where $i_1, i_0 \in \{1, \ldots, m\}$ is the finite set of values that the discrete-time random variable takes. This formula describes that the probability of the future states of the random variable depends only on the current state and not the ones that precede, which is referred to as the Markov property. For all combinations of $i_0$ and $i_1$, a transition matrix can be constructed that contains the conditional probabilities q of states at the next time step $i_1$, given the present state $i_0$ as transition probabilities. Each row of the matrix is a probability distribution summing to 1.

The order of the Markov process indicates the number of previous observations on which the next state of the variable statistically depends. In an $\ell$-th order MC model, the probability that the system will be in a particular state at time $t$ depends not only on its

state at time $t-1$ but also on the states at times $t-2; t-3; \ldots; t-\ell$. For example, a second order MC model can be expressed by:

$$P(X_t|X_0,\ldots,X_{t-2},X_{t-1}) = P(X_t|X_{t-2},X_{t-1}) = q_{i_2 i_1 i_0} \tag{9}$$

The transition probabilities can be estimated from the given set of observations by counting the number of transitions $n_{i_\ell \ldots i_0}$ and computing the conditional frequencies:

$$\hat{q}_{i_\ell \ldots i_0} = \frac{n_{i_\ell \ldots i_0}}{n_{i_\ell \ldots i_1 +}} \tag{10}$$

where

$$n_{i_\ell \ldots i_1 +} = \sum_{i_0=1}^{m} n_{i_\ell \ldots i_0} \tag{11}$$

$m$ is the number of discrete values of the random variable, and $\ell$ is the order of the MC.

Solar-radiation-derived CMF possesses the Markov property, which permits us to predict solar radiation via an MC model. By computing the autocorrelation of CMF for the training set in Berlin (Figure 2), we observe the significant serial correlation of CMF, which is frequently seen in atmospheric variables. The autocorrelation of lag-1 declines from about 0.9 to lower than 0.8 for lag-2 (30 min). These high values of autocorrelation show the high dependence of the current state of the system on the previous 2 time steps, which is a determent factor for the selection of the MC order (see Section 3.2.1). Regarding the probabilities q of the CMF transitions, we assume them to be time-invariant. However, this assumption may be affected by seasonal variations, as can be observed in the annual course of monthly mean CMF in Figure 1. The test results for the assumption of time homogeneity based on seasonal sub-sampling are provided in Appendix A.1. Since the climatological conditions of different regions are not identical, the seasonal division suited for Berlin might not be readily applicable to all other locations. To reduce the complexity and retain the generality of the analysis, we do not construct the MC on a seasonal basis, yet the seasonal perspective could be of interest to future studies in this direction. For example, the consideration of periodicity in non-homogeneous Markov systems [22] and sub-sampling by several terms within a year [23] are referred to in the literature.

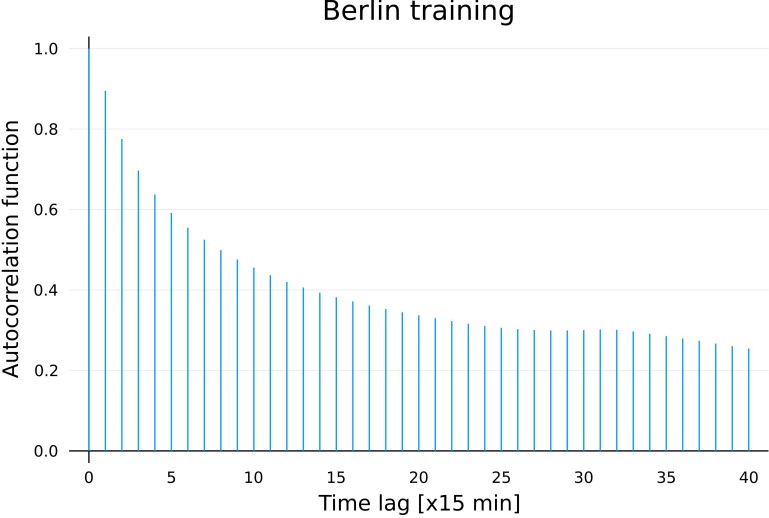

**Figure 2.** Autocorrelation of CMF with time lags from 0 to 40 time steps for the training set (2004–2018) in Berlin.

### 3.2.1. Categorization of CMF and Order Selection of MC

According to the definition of CMF in Section 2, it can take any value between 0 and 1. To apply CMF in the MC model, the continuous variable first needs to be discretized

into several classes. The next step involves setting the number of class m, which reflects a trade-off between the accuracy and complexity of the MC model. Figure 3 shows the standard deviation of the second order MC prediction bias for Berlin in 2019. It ranges from one to four time steps ahead in the year 2019, using different numbers for the CMF class, from 5 to 40 classes in an interval of 5. While 40 classes deliver even lower biases, we opt for 30 classes in order not to overfit the training set in case the classification generalizes unsatisfactorily, for example, by being too specific to the historical record to be applied to a later time.

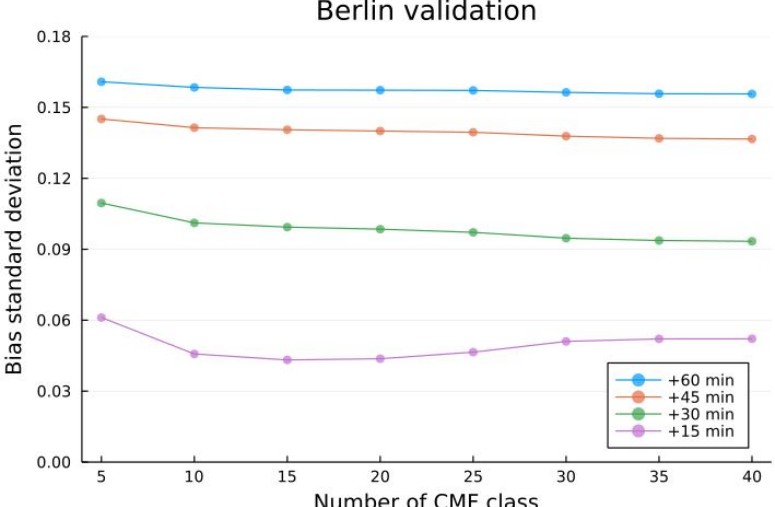

**Figure 3.** Bias standard deviation of MC-based model with different numbers of classes for the validation set (2019) in Berlin.

The categorization of CMF is implemented as follows: first, we categorize the training CMF values into 30 classes, each having the same number of data points. Consequently, the ranges of the CMF classes are not equal; there are larger (smaller) intervals for smaller (higher) CMF values towards 0 (towards 1). We compute the mean CMF value within each class, and these 30 representative values are used for the prediction in combination with the transition matrix. Figure 4 displays the distributions of each CMF class for the three Berlin data sets. The validation and test sets inherit their classifications from the training set; thus, they do not have an equal number of data points in every class. Classes with very high CMF values (clear sky) are in the majority throughout the whole period.

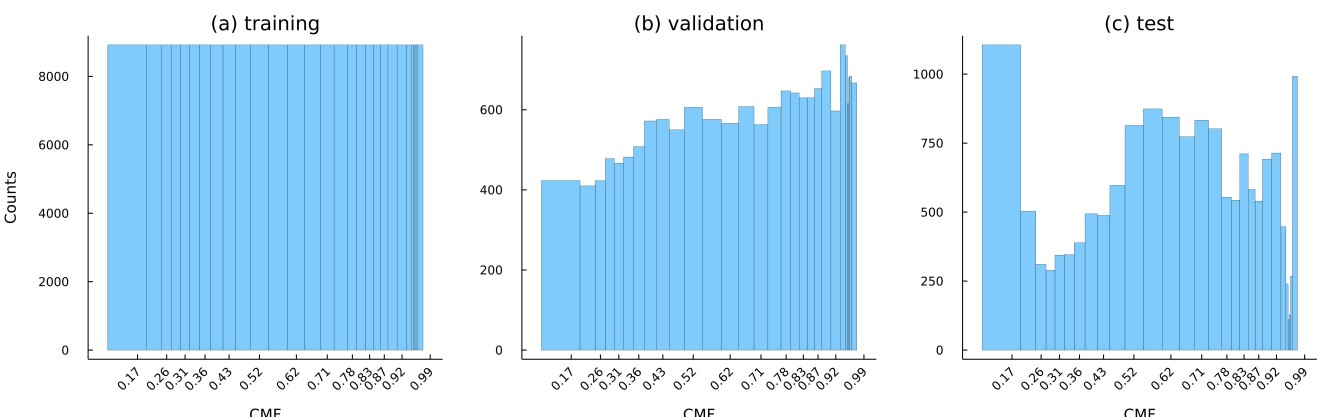

**Figure 4.** Distribution of CMF class for three data sets in Berlin: (**a**) training set for 2004–2018, (**b**) validation set for 2019, and (**c**) test set for 2020.

To evaluate the selection of the MC order $\ell$, we use the Akaike information criterion (AIC):

$$\text{AIC}(\ell) = -2\text{LL} + 2m^\ell(m-1) \tag{12}$$

where $m = 30$ and the log-likelihood of the time series LL is written:

$$\text{LL} = \sum_{i_\ell,\ldots,i_0=1}^{m} n_{i_\ell\ldots i_0}\ln(\hat{q}_{i_\ell\ldots i_0}) \tag{13}$$

Table 2 lists the AIC of a 30-class MC for $\ell \in \{1,2,3\}$. It can be seen that the AIC favors the selection of a second-order-MC-based model.

**Table 2.** AIC of a 30-class MC for $\ell \in \{1,2,3\}$ using the training set in Berlin.

| Order | AIC [$\times 10^6$] |
|-------|---------------------|
| 1 | 1.190 |
| 2 | 1.123 |
| 3 | 2.477 |

3.2.2. Temporal Extrapolation

Based on the results of AIC, we choose the order of the MC to be 2, and we distinguish 2 variants of predicting CMF values as follows.

*Variant a* From the validation or test set, we take 2 consecutive values as the observation and compute the value at the next step from the pre-calculated transition matrix. Rather than using a random sampling from a uniformly distributed variable, as elected in some previous studies such as [14,17], we compute the prediction value $\hat{X}$ by weighted mean at the specific row (transition) in the transition matrix, i.e., by summing up the product of the average of each CMF class $\overline{X}_i$ and the probability of each class $q_i$:

$$\hat{X} = \sum_{i_2 i_1 i_0=1}^{m=30} q_{i_2 i_1 i_0}\overline{X}_{i_2 i_1 i_0} \tag{14}$$

This predicted value is then set to be CMF at 1 to 4 time steps later. Figure 5 illustrates a diagram discerning two variants of temporal extrapolation during a one-hour time window from 9:00 to 10:00 in the MC-based model, for which the upper part specifies Variant a. Figure 6 visualizes the prediction of the second-order-MC-based model using this variant for the validation set in Berlin, with the horizontal axis representing the CMF class from 2 time steps earlier, and the vertical axis 1 time step earlier. The color in each cell denotes the CMF value for the time step to be predicted. The CMF value at the current step closely follows the range from CMF 2 time steps earlier, which indicates the memory of the system we take advantage of with the MC.

*Variant b* To predict CMF values more than 2 time steps ahead of the current time step, we take 2 consecutive values from the record as the observation, and we predict the value for the next time step following Variant a. For the prediction of one further time step, the previous 2 observation points consist of the latter one from the record and the newly predicted one, and we apply the prediction following Variant a. Therefore, for one step ahead, the results of both variants are the same, but only Variant a would be visualized to avoid redundancy and/or overlapping in the figures. If the value at one further time step is needed, the new observation would consist of the 2 consecutive predictions. We repeat this procedure until we reach the prediction for 4 time steps ahead. In Variant b, for a time window of one hour, only the first two CMF values are taken from the record, and the following inputs are either a combination of record and prediction or solely predicted values (see the lower part of Figure 5).

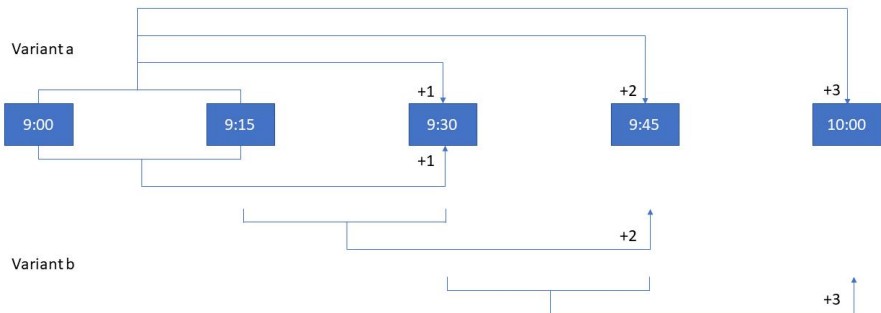

**Figure 5.** Diagram of two variants of temporal extrapolation during a one-hour time window from 9:00 to 10:00 in the MC-based model.

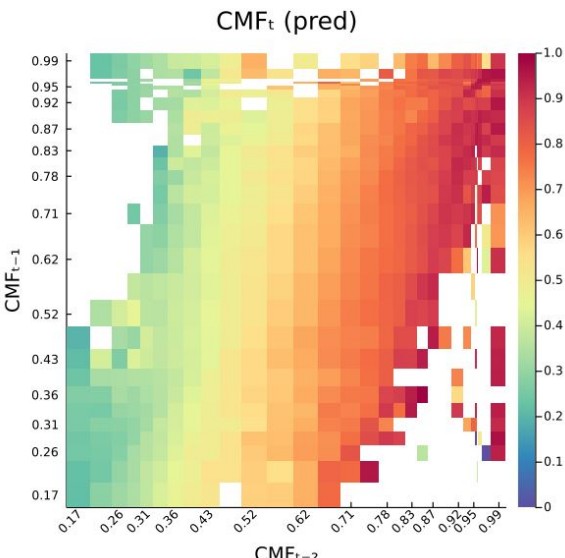

**Figure 6.** Prediction of the second-order-MC-based model for the validation set (2019) in Berlin.

### 3.3. Persistence Approach

The persistence approach assumes that the cloud state does not change from the previous to the current time step; therefore, it uses the CMF value for the previous time step as the prediction for the current time step. It is essentially a temporal shift of the original time series by a certain interval, in this case, from 1 to 4 time steps earlier to the present.

### 3.4. Neighbor Inference Approach

To trace the cloud movement above a region, one option is to look up the record of its neighboring region with a temporal lag. Using Berlin as a test bed, we evaluate the viability of inferring CMF values for the central grid cell from its neighbor cells at the previous time step. The neighbor cells are the central cell shifted to its eight directions by 0.1° longitude or/and latitude: to the west, northwest, north, northeast, east, southeast, south, and southwest. We first exclude the time steps with CMF values higher than or equal to 0.95 (sunny with very few clouds) and compute the Pearson correlation of the CMF values for the central cell with those for the neighbor cells to obtain eight correlation coefficients. Figure 7 displays the correlation coefficients of the CMF values in Berlin with CMF values from the previous time step (15 min prior) in the eight neighbor cells, based on the time series from 2004 to 2019. The highest correlation is with the cell to the west, followed by the cells in the northwest and southwest. Moving towards the east, the correlation decreases, and the minimum correlation is situated in the southeastern cell. This is as expected, since the general movement of clouds above Berlin is a westerly flow. In the following section, we evaluate the performance of using prior CMF values from the western cell in comparison to other prediction techniques.

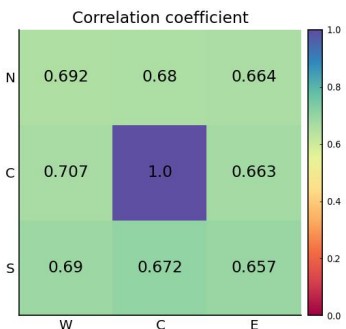

**Figure 7.** Correlation coefficients of CMF in Berlin with its eight neighboring grid cells.

### 3.5. Hybrid Approach

For the 2019 time series, we evaluate the accuracy of the persistence method, neighbor inference, and MC predictions including both Variant a and Variant b, in order to develop hybrid methods based on MAE or RMSE for all 30 CMF classes. MAE measures the mean difference between predicted and actual time series, whereas RMSE measures the bias and standard deviation of the difference between predicted and actual time series, and both have the same unit as the actual time series (dimensionless in the case of CMF). If, for a given CMF class, the error rate of a certain approach is the lowest among the considered approaches, we adopt that approach for the CMF class in question. We then apply this to the 2020 time series by choosing the optimal approach for the next time step based on the CMF class at the current time step. Table 3 gives an overview of the different approaches and their respective indices in Section 4. A visualization of the hybrid approach can be found in Figure 8.

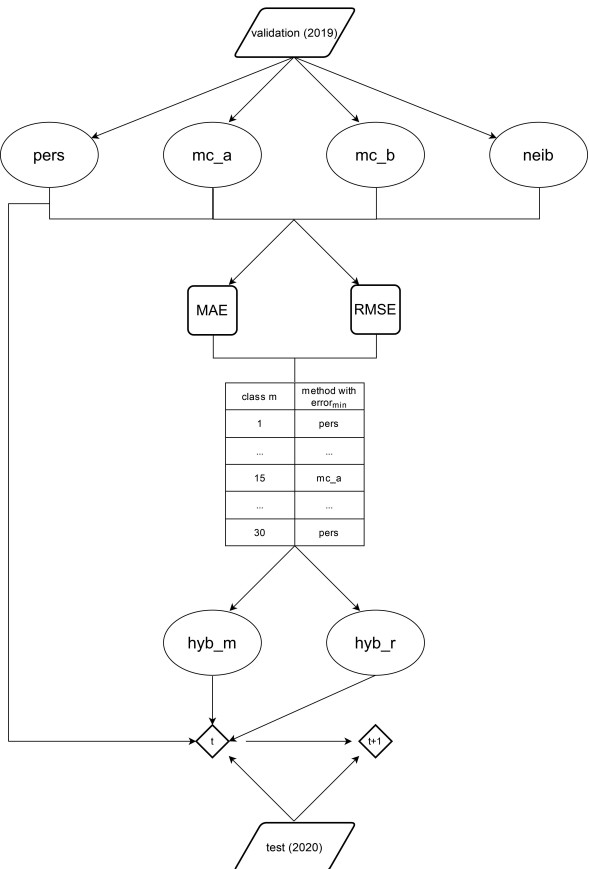

**Figure 8.** Flux diagram illustrating the hybrid approach. The indices of the methods can be found in Table 3.

**Table 3.** Overview of prediction methods with their indices and color codes.

| Method | Index | Color |
|---|---|---|
| Persistence | pers | violet |
| MC Variant a | mc_a | sky blue |
| MC Variant b | mc_b | pink |
| Neighbor inference | neib | olive |
| Hybrid by MAE | hyb_m | black |
| Hybrid by RMSE | hyb_r | red |

*3.6. GHI Prediction*

For the computation of all-sky GHI, we multiply clear-sky GHI provided by CAMS by the predicted CMF values following different approaches:

$$GHI = GHI_{CS} \cdot CMF \tag{15}$$

## 4. Results and Discussion

*4.1. Results of CMF*

In this section, we first present the results for the evaluation of CMF for Berlin and then for the other three cities. Table 4 lists, for the different methods, the Pearson correlation coefficients of predicted time series with the actual time series for one to four time steps ahead in 2020 for Berlin. Within each lead time step, the correlation coefficients between the prediction and the record are similar among different methods, ranging from around 0.9 for one time step ahead to approximately 0.7 for four time steps ahead. The MC-based model Variant a outperforms the persistence approach for a forecast horizon of within half an hour. For a lead time of longer than 45 min, neighbor inference wins out and has a higher correlation with the actual time series than the persistence or MC predictions. Built upon these comparisons, the hybrid approaches outperform the other methods; the method based on RMSE is slightly better than the one based on MAE, except in the case of a lead time of one hour.

**Table 4.** Correlation coefficients of predicted and actual CMF time series for four lead time steps (+1 to +4) in 2020 for Berlin. The results for the other three cities can be found in Tables A1–A3 in Appendix A.

| Method | +1 | +2 | +3 | +4 |
|---|---|---|---|---|
| pers | 0.9087 | 0.8142 | 0.7635 | 0.7275 |
| mc_a | 0.914 | 0.8204 | 0.7658 | 0.7254 |
| mc_b | 0.914 | 0.7956 | 0.72 | 0.6691 |
| neib | 0.8874 | 0.805 | 0.7653 | 0.7331 |
| hyb_m | 0.9143 | 0.8301 | 0.7946 | 0.7628 |
| hyb_r | 0.9146 | 0.8352 | 0.7975 | 0.7609 |

Next, we examine the agreement of the CMF values between different methods and the actual time series, using the standard deviation of the prediction bias. Figure 9 shows the standard deviation of bias for CMF from different approaches. In general, the standard deviation of the prediction bias for each approach increases with the lead time. For one time step ahead, the standard deviations of bias for the hybrid approaches are very close to that for the persistence approach. As the lead time increases, the differences between them also increase. That is, the standard deviation of bias for the persistence method increases more than for hybrid approaches. When the lead time is 15 min, the standard deviation for hybrid approach based on RMSE is almost equal to that based on MAE; however, in later time steps, the ones based on RMSE have slightly lower standard deviations than the ones based on MAE.

In the first and second lead time step, the neighbor inference approach has the highest standard deviation of bias among the methods considered. For three and four time steps

ahead, the standard deviation of bias becomes slightly smaller than that for the persistence method. The MC Variant a prediction method has a lower standard deviation of bias throughout than the persistence approach. For one time step ahead, the standard deviation for the MC Variant a prediction method is almost equal to that of the hybrid approaches. On the other hand, the MC Variant b prediction method has a relatively large standard deviation for 15 to 60 min of lead time. It only performs a little better than the neighboring grid cell method for two time steps ahead, but has the highest standard deviation among all methods for lead times of more than 30 min. For 1 h ahead, the standard deviation for the MC Variant b prediction method exceeds 0.2.

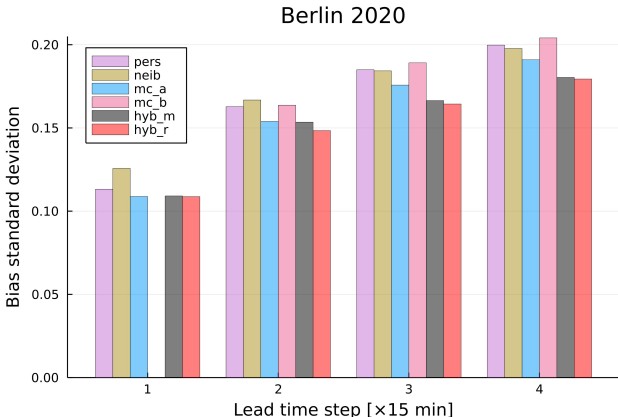

**Figure 9.** Standard deviation of CMF prediction bias for Berlin. The results for the other three cities can be found in Figure A2 in Appendix A.

In addition to the comparison between the prediction and the actual time series, we also compare the errors in prediction among the different methods by means of MAE and RMSE. Figure 10 shows the MAE and RMSE of CMF from different methods for four lead time steps in 2020 for Berlin, from the lower left to upper right corner. For 15 min ahead, the error rates of the hybrid methods are close to that of the persistence method. From 30 min ahead onward, the error rates of the hybrid methods are notably lower than that of the persistence method. The error rates of the persistence method and the neighbor inference approach for longer than 45 min ahead are even larger than those of the hybrid methods for one hour ahead. While the MAEs of the MC Variant a prediction method are, for all lead time steps, larger than those of the other methods except the MC Variant b prediction method, its RMSE is comparable to the others for one time step ahead and is evidently smaller than those of the persistence or neighboring approaches from 30 min ahead onward. On the other hand, the MC Variant b prediction method has no advantage over the other methods, regardless of MAE or RMSE across the three lead time steps analyzed.

The averaging of errors could mask the variations in forecast performance due to cloud conditions [24]. Thus, we look into the more informative prediction error for each CMF class. Therein, we compute the MAE of each CMF class by the persistence and hybrid approaches for one to four time steps ahead (Figure 11). The MAE of each CMF class grows as the lead time increases, as expected. For 15 to 30 min ahead, the persistence method has an advantage in the CMF range between 0.3 and 0.4. This arises due to the fact that the hybrid methods consider the prediction errors of the immediately previous one time step, the CMF class of which could be different from the current time step. For further lead time steps, it is almost invariably the hybrid approaches that generate lower error rates across the whole CMF range. The improvement offered by the hybrid methods is the most evident for the CMF classes from 0.5 to 0.8, especially for a lead time of one hour. Except for one time step ahead, the error rates are high for very small CMF classes. For CMF classes of around 0.8 to 0.9, there is another high MAE peak, hinting at the difficulty of accurately predicting the cloud state at these levels. In the case of Berlin, the hybrid

approach based on RMSE is more accurate than that based on MAE, except for a few CMF classes at around 0.83.

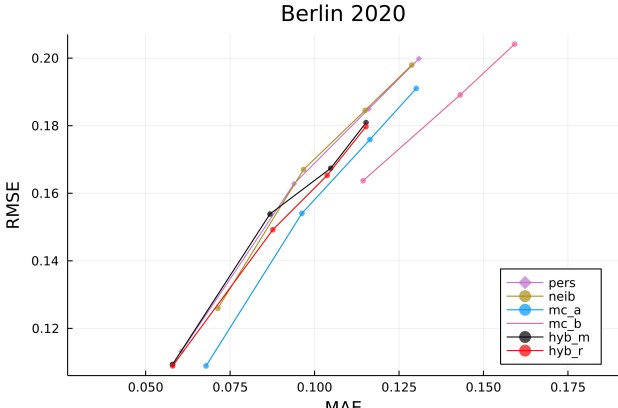

**Figure 10.** Mean absolute error (MAE) versus root mean square error (RMSE) for Berlin. For one step ahead (+1), the results for both $mc_a$ and $mc_b$ are the same, but only Variant a is visualized to avoid redundancy and/or overlapping. The same applies to the following figures. The results for the other three cities can be found in Figure A3 in Appendix A.

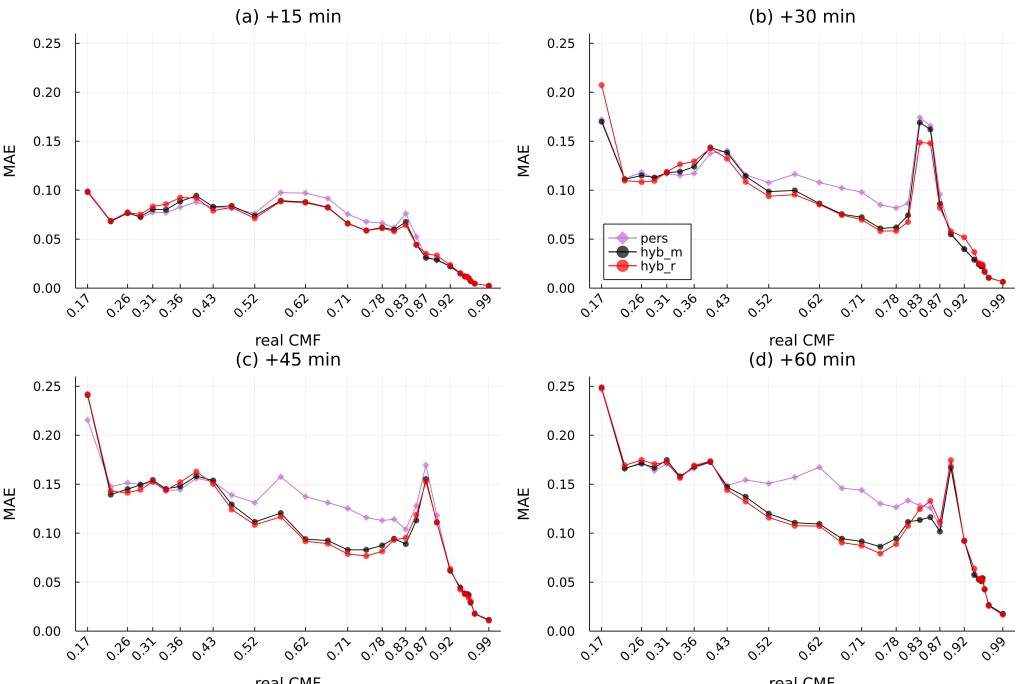

**Figure 11.** MAE for each CMF class for (**a**) 15 min, (**b**) 30 min, (**c**) 45 min, and (**d**) 60 min ahead in 2020 for Berlin. The results for the other three cities can be found in Figures A4–A6 in Appendix A.

We identify the effectiveness of the proposed approaches in a retrospective way. The applicability of these methods mainly depends on current, known information. The change in CMF is the difference in CMF between the current and the immediately previous time step(s), which can be considered known information. For example, if we were currently at 9:30, the CMF change from 9:00 ($t-2$) to 9:30 ($t$) would be known. Thus, we evaluate the MAE of predicted CMF as a function of each class of CMF change using different approaches for one to four time steps ahead (Figure 12). The histograms in the background indicate the number of data points in each class of CMF that change from the preceding time step. The persistence method performs best when the change in CMF is very small, within the range of ±0.04. When the CMF change is larger, the majority of the other approaches

outperform the persistence method. Among them, Variant b of the MC prediction method has the lowest MAE when the CMF changes considerably, followed by Variant a or the hybrid approach, depending on the class of CMF change. These tendencies are more distinct when the CMF increases from the past to the present than when the CMF change is in the reverse direction. However, note that the majority of CMF changes reside in a very small range clustered around zero, as can be seen from the distribution of the CMF changes.

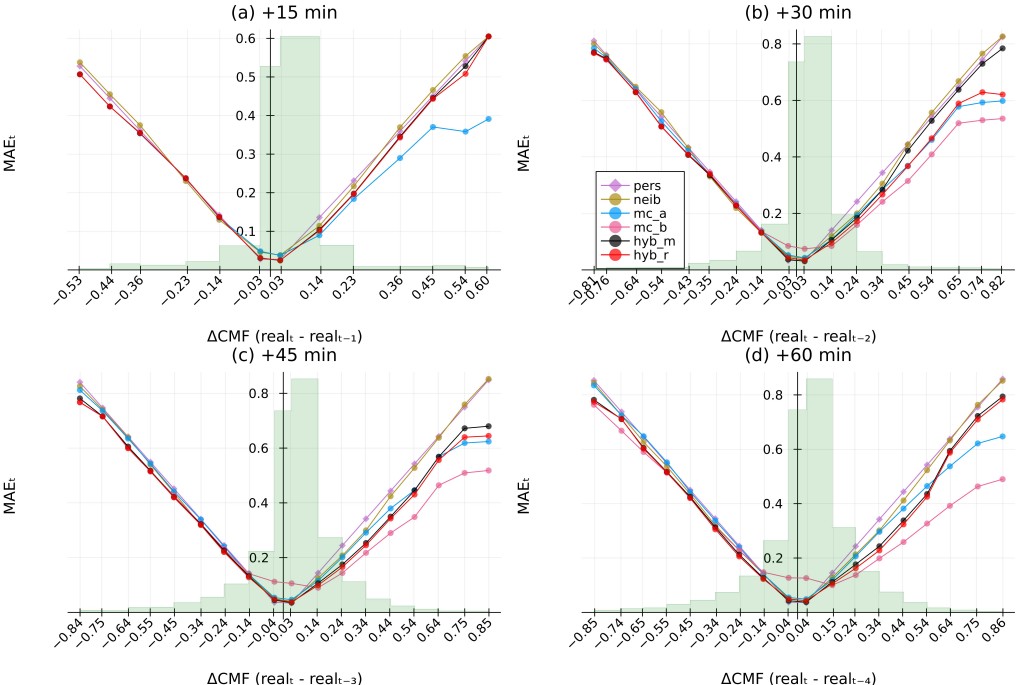

**Figure 12.** MAE of CMF change for (**a**) 15 min, (**b**) 30 min, (**c**) 45 min, and (**d**) 60 min ahead for Berlin. The horizontal ticks are the means of every class of CMF change, where the change classes are equally spaced with an interval of 0.1, e.g., [−0.1, 0) and [0, 0.1). The results for the other three cities can be found in Figures A7–A9 in Appendix A.

After the evaluation of CMF for the city of Berlin, we apply the hybrid method based on RMSE to the three other cities introduced in Section 2. The individual transition matrix is computed for each city. Table 5 lists the Pearson correlation coefficients of CMF predicted by the hybrid method based on RMSE with the actual time series, for four lead time steps for four cities in 2020. The correlation is highest for one time step ahead, featuring correlation coefficients larger than 0.9. As the lead time increases, the correlation coefficients decrease to around 0.75. Among the cities considered, the hybrid approach yields the best result for Bucharest for each lead time step, even though Bucharest is not the city with the lowest or the most variable CMF (see Table 1). For Helsinki, the hybrid method works properly from one to four time steps ahead. In the sunnier city Athens, the hybrid method's prediction is comparably good for at least 15 or 30 min ahead.

**Table 5.** Correlation coefficients of predicted CMF by hyb_r with the actual time series for four cities for four lead time steps (+1 to +4) in 2020.

| City | +1 | +2 | +3 | +4 |
|---|---|---|---|---|
| Athens | 0.9176 | 0.8406 | 0.7807 | 0.7574 |
| Bucharest | 0.9547 | 0.8892 | 0.8365 | 0.7962 |
| Berlin | 0.9146 | 0.8352 | 0.7975 | 0.7609 |
| Helsinki | 0.9418 | 0.8536 | 0.81 | 0.7696 |

### 4.2. Results of GHI

Following the method described in Section 3.6, we examine the performance of the hybrid approach based on RMSE in predicting GHI. Figure 13 shows the GHI time series provided by CAMS and predicted with a lead time of 15 min using the hybrid approach based on RMSE for Berlin from 1 January 2020 to 11 January 2020. Nighttime hours without sunshine are not included. We are able to observe both curves in high agreement, especially when the absolute GHI values are high. In moments of relatively low GHI values, the hybrid approach based on RMSE tends to overestimate the GHI, as illustrated by the red overshoots during the course.

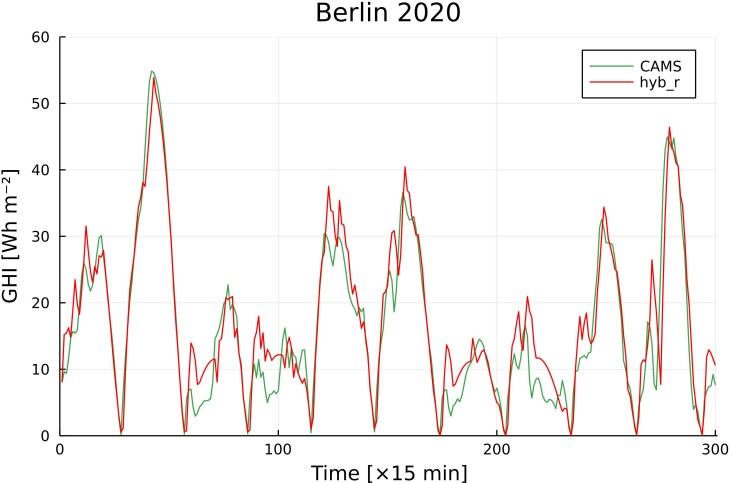

**Figure 13.** Comparison of actual daily GHI by CAMS (green) and predicted with a lead time of 15 min by the hybrid approach based on RMSE (red) for Berlin from 1 January 2020 to 11 January 2020.

When we consider the mean daily insolation (GHI) for each month, the annual course of the prediction for Berlin in 2020 bears a reasonably high resemblance to the one provided by CAMS (Figure 14), since shorter temporal variations of GHI are smoothed out. Especially in the months from July to September, the differences between both time series are negligible. In some months, where there is a visible deviation of monthly GHI prediction from the actual time series, there is still an overestimation from the hybrid method, except in April. The positive bias stemming from the MC-based model could be partially due to the way the model is trained in this study: the calculation of GHI or CMF does not take into account the atmospheric mechanisms that could affect cloudiness.

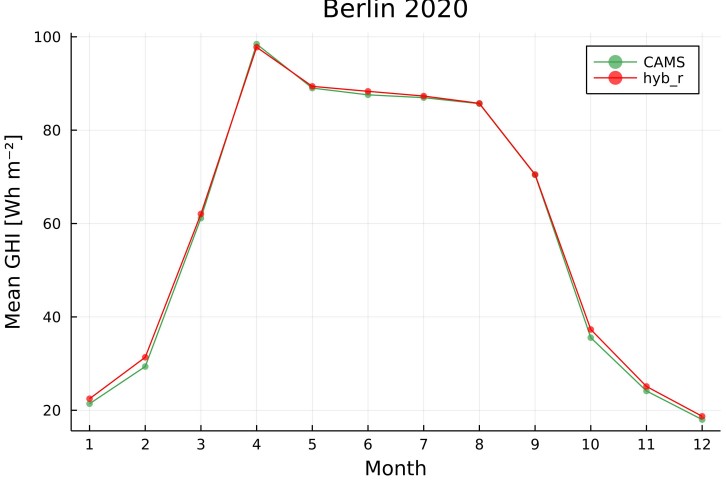

**Figure 14.** Comparison of actual monthly GHI by CAMS (green) and predicted with a lead time of 15 min by the hybrid approach based on RMSE (red) for Berlin in 2020.

Figure 15 shows the monthly MAE of GHI 1 h ahead, as predicted by the persistence and hybrid methods for the four cities in 2020 with their individual vertical axes. In Athens, the performance of the persistence method is not necessarily worse than either hybrid method, except from October to February, a time of year when the daily sunshine duration is shorter. However, the hybrid approach based on MAE has the lowest MAE in every month of the year. In Bucharest, both hybrid methods perform similarly, almost always better than the persistence method, especially in summer from May to July. The advantage of the hybrid approach over the persistence method can be better demonstrated in the city of Berlin, particularly during summer, with a monthly MAE reduction of up to 2.5 Wh m$^{-2}$. In Helsinki, the improvement offered by the hybrid approaches in prediction accuracy is more evident in the last quarter of the year, from October to December.

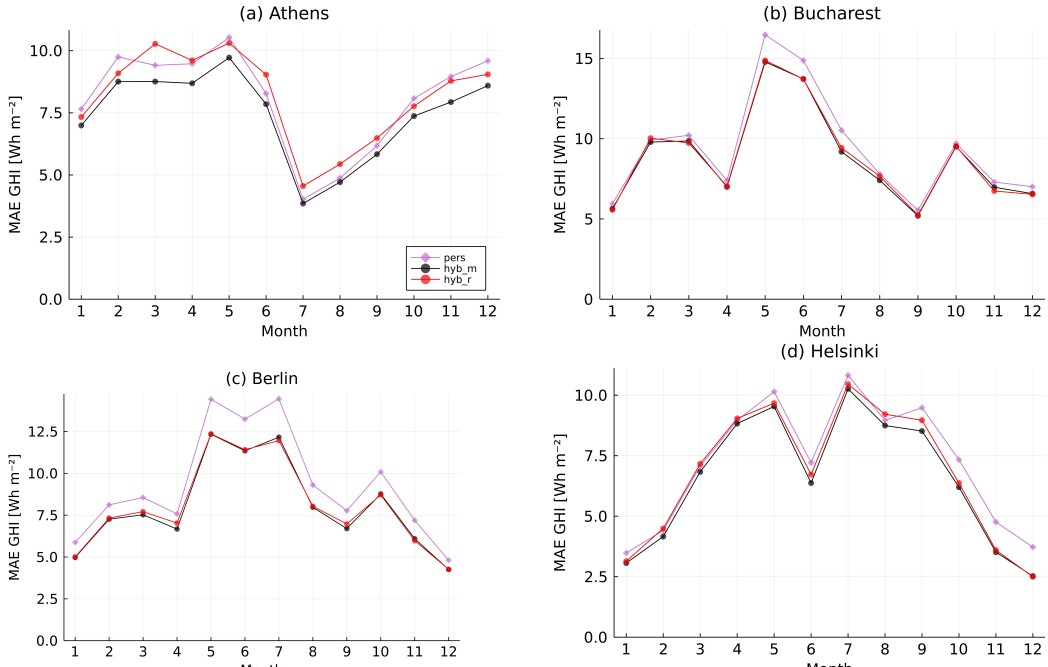

**Figure 15.** MAE of 1 h ahead GHI for the persistence and hybrid approaches in 2020 for (**a**) Athens, (**b**) Bucharest, (**c**) Berlin, and (**d**) Helsinki.

To put the results of GHI prediction into perspective, we compare the performance of the hybrid methods with the persistence method using the MAE improvement of GHI for each month. This improvement in MAE (in %) is defined as the difference in MAE of GHI by the hybrid approach (MAE($\hat{X}_i$)) and MAE of GHI by the persistence approach (MAE($X_i$)) divided by the actual mean daily GHI in Month $i$ ($\overline{X}_i$):

$$\Delta\text{MAE}_{i,rel} = \frac{\text{MAE}(\hat{X}_i) - \text{MAE}(X_i)}{\overline{X}_i} \tag{16}$$

Figure 16 shows the monthly relative improvement of the hybrid approach based on MAE compared with the persistence method for four time steps for the four cities. Note that the vertical axis scales have also been individually adjusted to make the results more readable. For one time step ahead in Athens, the hybrid approach is inferior to the persistence approach in predicting GHI. There is some improvement for further time steps, although the magnitude is small. The improvement in most of the months is within 2%, and is close to zero in the summer months of July and August. In Bucharest, there is also no improvement for one time step ahead except for the months from May to July, with small magnitude. The improvement for the majority of months is smaller than 2%. In Berlin, on the other hand, we observe an improvement from one to four time steps ahead across all months, except for 15 min ahead in February. For a lead time shorter than 45 min,

the fluctuation of improvements throughout the year is relatively small. However, when predicting one hour ahead, the improvement is apparent during the winter months of October to January, even reaching nearly 5% in November. In Helsinki, there is a general improvement confirmed. In December, we even observe an improvement higher than 10% for the lead times of 45 and 60 min. However, this is mainly due to the original low monthly GHI in Helsinki.

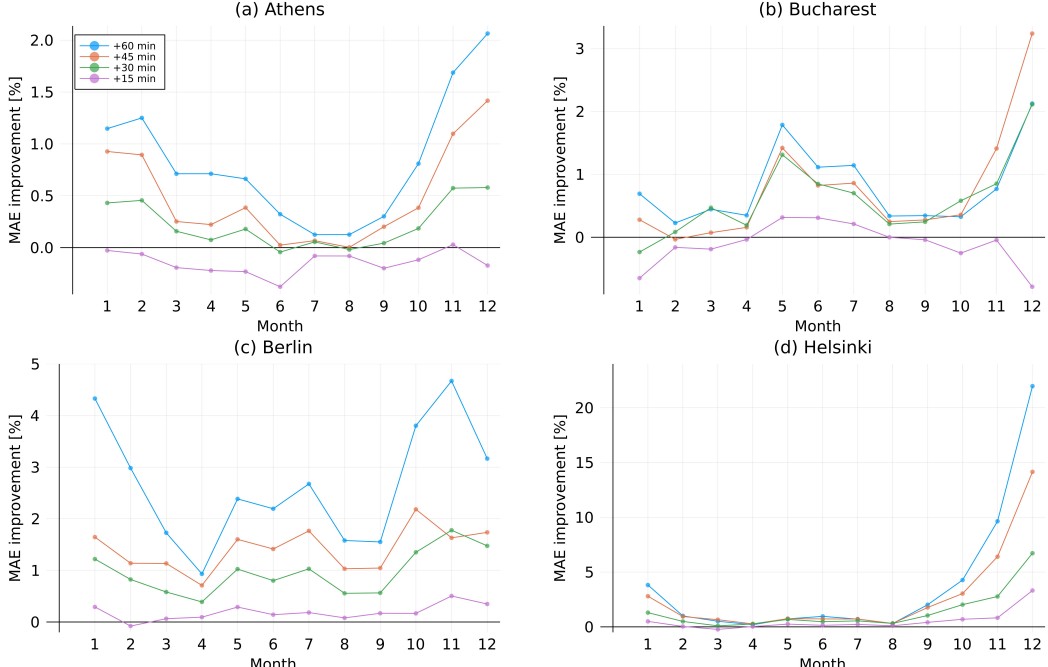

**Figure 16.** Relative improvement of the hybrid approach in 2020 for (**a**) Athens, (**b**) Bucharest, (**c**) Berlin, and (**d**) Helsinki.

## 5. Conclusions

The accuracy of solar radiation prediction is subject to the location and forecast horizon considered. For locations with relatively high variability in cloudiness, such as Berlin in this study, the persistence approach does not work satisfactorily for a longer period of time. This is linked to our motivation to offer better methodologies for solar nowcasting. In this study, we present a hybrid model for solar radiation nowcasting that combines the persistence ensemble, the Markov chain model, and neighbor inference through cloud movement. In the GHI prediction for Berlin, the reduction in monthly mean error amounts to 2.5 Wh m$^{-2}$, and the relative improvement reaches nearly 5% compared with the traditional persistence method. In the other three cities, accuracy improvements of various degrees can also be observed.

We acknowledge that the data involved in this study are based on outputs from models rather than true measurements, which implies limitations on the ability to 100% reflect reality. However, note that the proposed model undoubtedly could also be applied to ground-based measurement data. On the other hand, the use of modeled clear-sky radiation is inevitable for the prediction of GHI from CMF even outside modeling simulation involvements.

The presented approach has the advantage that its straightforward model configuration does not require the input of other variables, such as wind speed or CMV from all-sky imagers, both of which can introduce multiple uncertainties. Furthermore, the computational time required for prediction at one single site is trivial. Our approach also adjusts the transition matrix according to the input data for the MC prediction; thus, it is a generic model that can be readily applied to different geographical locations where the CAMS data are available. Therefore, one prospect for the hybrid model is to upscale its ap-

plication to, e.g., the pan-European domain to build an efficient network for solar radiation nowcasting. The presented method can be the base for, or part of, hybrid approaches that include additional, satellite-based cloud information and/or ground measurements (cloud cameras, solar irradiance instruments, and other atmospheric parameters measurements) with the goal of improving solar forecasting.

**Author Contributions:** Conceptualization and methodology, S.K., X.H. and K.P.; software and formal analysis, X.H. and K.P.; data curation, visualization, validation, and writing—original draft preparation, X.H.; writing—review and editing, X.H., K.P., Y.-M.S.-D. and S.K.; supervision and funding acquisition, S.K. All authors have read and agreed to the published version of the manuscript.

**Funding:** This research was funded by "European Commission EuroGEO Showcases: Applications Powered by Europe" project (grant no. 820852).

**Acknowledgments:** Data from CAMS radiation service are accessed through Transvalor SoDA at http://www.soda-pro.com/web-services/radiation/cams-radiation-service (accessed on 10 November 2021).

**Conflicts of Interest:** The authors declare no conflict of interest. The funders had no role in the design of the study; in the collection, analyses, or interpretation of data; in the writing of the manuscript, or in the decision to publish the results.

## Appendix A

**Table A1.** Correlation coefficients of predicted and actual cloud modification factor (CMF) time series for 4 lead time steps (+1 to +4) in 2020 for Athens.

| Method | +1 | +2 | +3 | +4 |
|---|---|---|---|---|
| pers | 0.9157 | 0.8323 | 0.7867 | 0.7509 |
| mc_a | 0.9164 | 0.8394 | 0.7929 | 0.7587 |
| mc_b | 0.9164 | 0.816 | 0.7498 | 0.6981 |
| neib | 0.8332 | 0.7626 | 0.7329 | 0.7065 |
| hyb_m | 0.9174 | 0.8475 | 0.8069 | 0.7821 |
| hyb_r | 0.9176 | 0.8406 | 0.7807 | 0.7574 |

**Table A2.** Correlation coefficients of predicted and actual CMF time series for 4 lead time steps (+1 to +4) in 2020 for Bucharest.

| Method | +1 | +2 | +3 | +4 |
|---|---|---|---|---|
| pers | 0.9226 | 0.8457 | 0.8023 | 0.7663 |
| mc_a | 0.955 | 0.8811 | 0.8281 | 0.7854 |
| mc_b | 0.955 | 0.885 | 0.8136 | 0.7549 |
| neib | 0.8953 | 0.8281 | 0.7943 | 0.7631 |
| hyb_m | 0.9547 | 0.8919 | 0.8381 | 0.7954 |
| hyb_r | 0.9547 | 0.8892 | 0.8365 | 0.7962 |

**Table A3.** Correlation coefficients of predicted and actual CMF time series for 4 lead time steps (+1 to +4) in 2020 for Helsinki.

| Method | +1 | +2 | +3 | +4 |
|---|---|---|---|---|
| pers | 0.8974 | 0.7858 | 0.7319 | 0.6943 |
| mc_a | 0.9424 | 0.8408 | 0.7715 | 0.7189 |
| mc_b | 0.9424 | 0.8413 | 0.7488 | 0.6651 |
| neib | 0.8878 | 0.7831 | 0.7342 | 0.6982 |
| hyb_m | 0.941 | 0.8551 | 0.8094 | 0.7725 |
| hyb_r | 0.9418 | 0.8536 | 0.81 | 0.7696 |

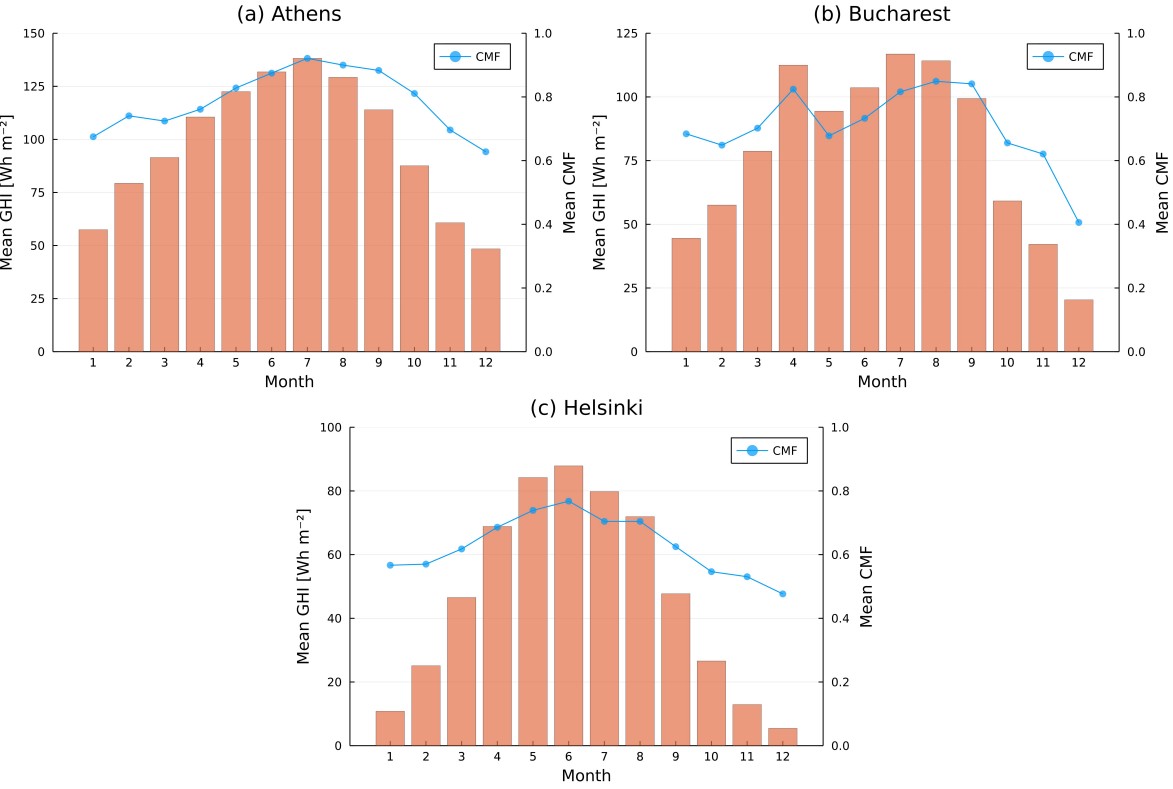

**Figure A1.** Daily mean global horizontal irradiation (GHI) for each month (orange bars, left axis) and monthly mean cloud modification factor (CMF; blue curves, right axis) in 2020 for (**a**) Athens, (**b**) Bucharest, and (**c**) Helsinki.

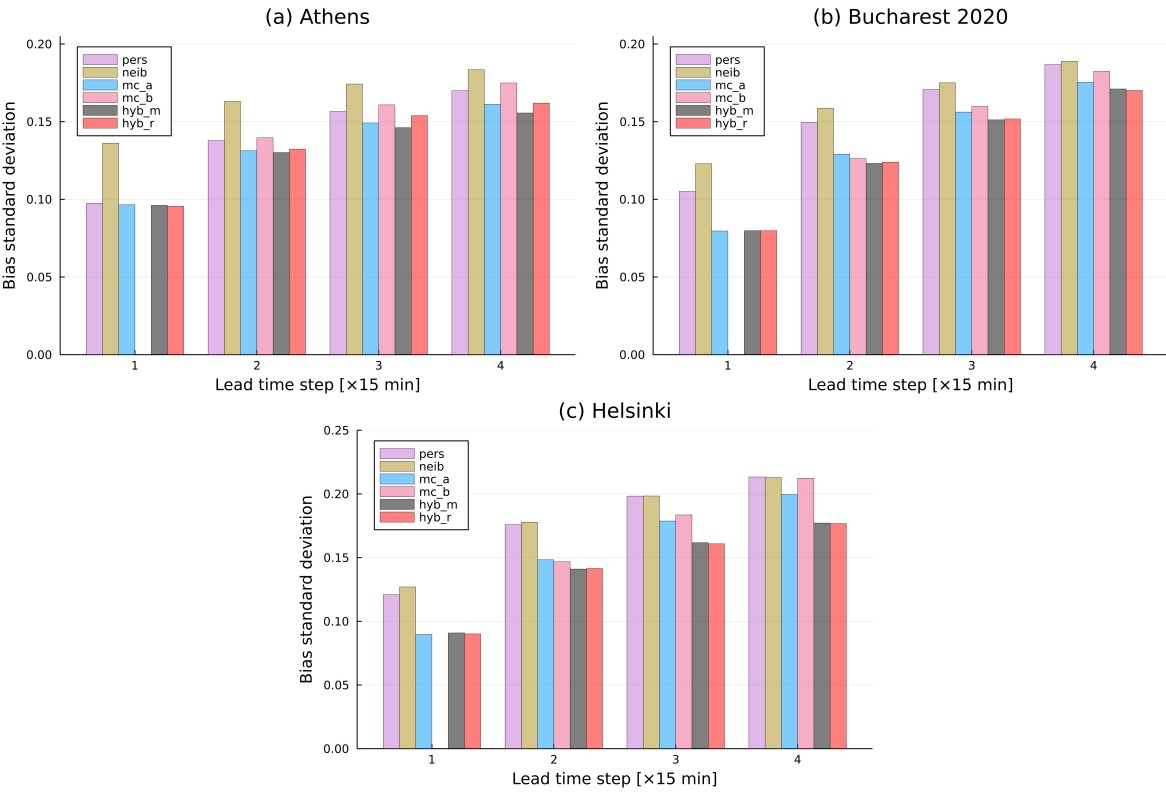

**Figure A2.** Standard deviation of cloud modification factor (CMF) prediction bias for (**a**) Athens, (**b**) Bucharest, and (**c**) Helsinki.

*Appendix A.1. Test for Time Homogeneity in the Transition Probabilities*

Let $n_{i_1 i_0}$ denote the number of transitions from a given state $i_0$ to state $i_1$ in the interval of observation $[0, T]$; the transition probabilities are

$$\hat{q}_{i_1 i_0} = \frac{n_{i_1 i_0}}{n_{i_1+}} \; for \; i_1, i_0 \in \{1, \dots, m\} \tag{A1}$$

with

$$n_{i_1+} = \sum_{i_0=1}^{m} n_{i_1 i_0} \tag{A2}$$

Using $m = 10$, we test the first order Markov chain with 10 states, formulating the null hypothesis as the following [25]:

$$\mathbb{H}_0 : \hat{q}_{i_1 i_0}(t) = \hat{q}_{i_1 i_0} \; for \; all \; i_1 \in F(i_0) \; and \; t, \; given \; i_0 \tag{A3}$$

where $\hat{q}_{i_1 i_0}$ denotes the time-independent transition probabilities and $\hat{q}_{i_1 i_0}(t)$ the time-dependent ones, and $F(i_0)$ is the set of values of $i_1$ for which $\hat{q}_{i_1 i_0}(t) > 0$.

Under $\mathbb{H}_0$ the statistic

$$\chi^2(i_0) = \sum_{t=0}^{T-1} \sum_{F(i_0)} n_{i_1+}(t) \frac{[\hat{q}_{i_0 i_1}(t) - \hat{q}_{i_0 i_1}]^2}{\hat{q}_{i_0 i_1}} \tag{A4}$$

is approximately $\chi^2$ distributed with $(T-1) \times [N(i_0) - 1]$ degrees of freedom, where $N(i_0)$ is the number of members of the set $F(i_0)$.

We divide the time series of the training set from 2004 to 2018 in Berlin into 4 three-month seasons, with the first season $t = 1$ consisting of December, January, February; $t = 2$ for the second season from March to May; and so on. The results of the calculated statistics for the four seasons are listed in Table A4.

**Table A4.** Test results for time homogeneity of the transition probabilities of the first order Markov chain with 10 states. The columns from left to right list the season, the sample size in each season, the degree of freedom, and the statistics, respectively.

| $t$ | Sample Size | DoF | Statistic |
|---|---|---|---|
| 1 | 45,345 | 97 | 4383.091 |
| 2 | 76,656 | 99 | 98.019 |
| 3 | 87,403 | 100 | 422.324 |
| 4 | 58,203 | 89 | 200.771 |
| **sum** | 267,607 | 385 | 5104.206 |

Comparing the statistics with the percentage points of the $\chi^2$ distribution, we can reject $\mathbb{H}_0$ with a significance level of 5% ($p = 0.05$), thus confirming that time inhomogeneity is present for the training data set from 2004 to 2018 in Berlin. Note here the class imbalance and the distinguished different statistics among the sub-samples, especially between winter and summer.

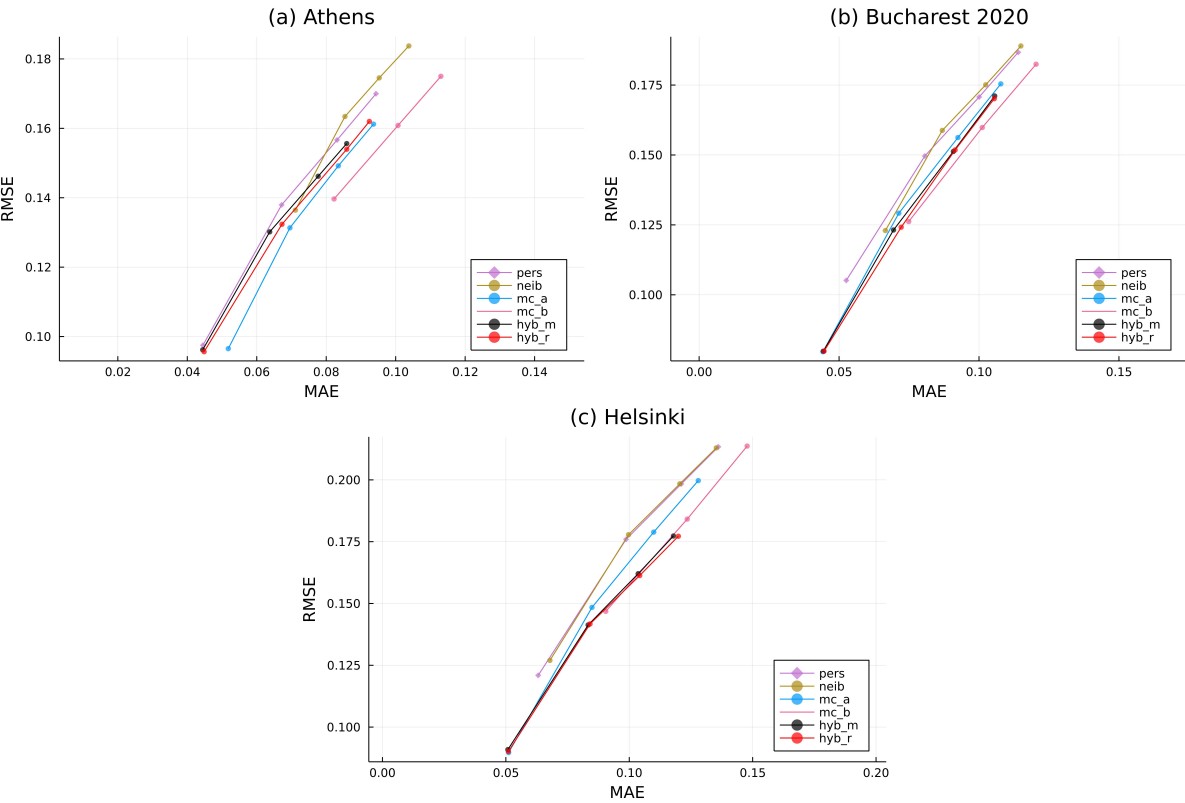

**Figure A3.** Mean absolute error (MAE) verses root mean square error (RMSE) for (**a**) Athens, (**b**) Bucharest, and (**c**) Helsinki.

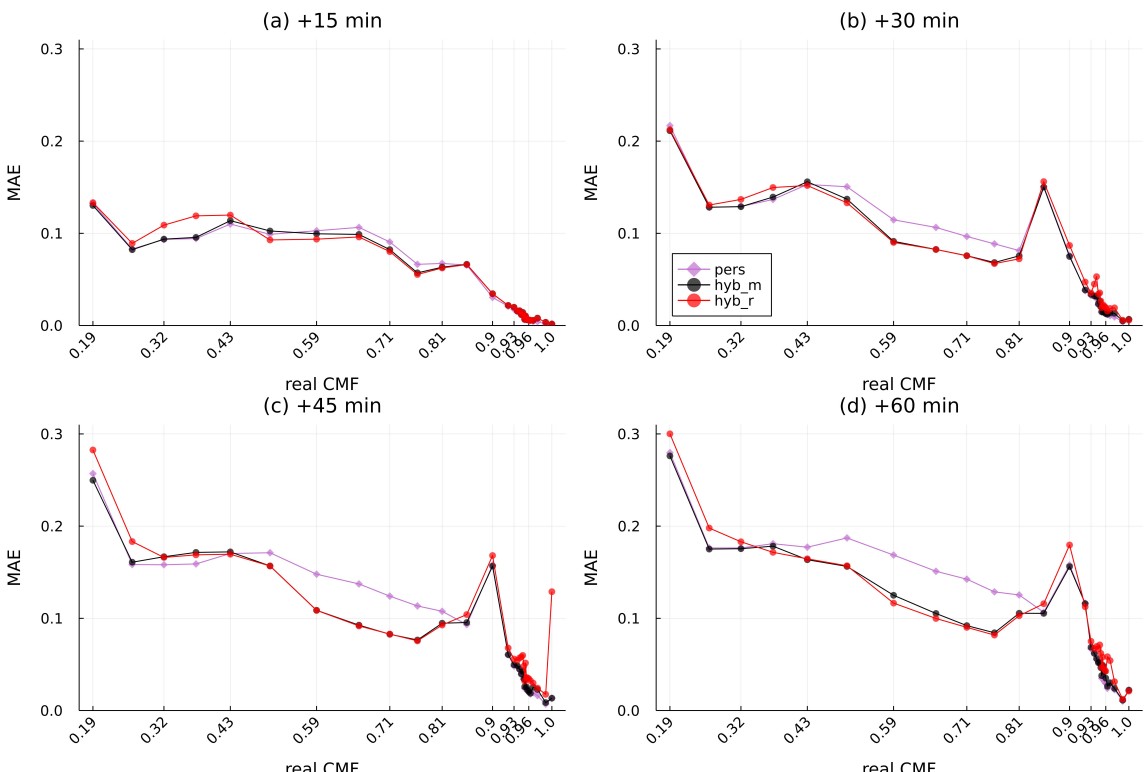

**Figure A4.** MAE for each CMF class for (**a**) 15 min, (**b**) 30 min, (**c**) 45 min, and (**d**) 60 min ahead in 2020 for Athens.

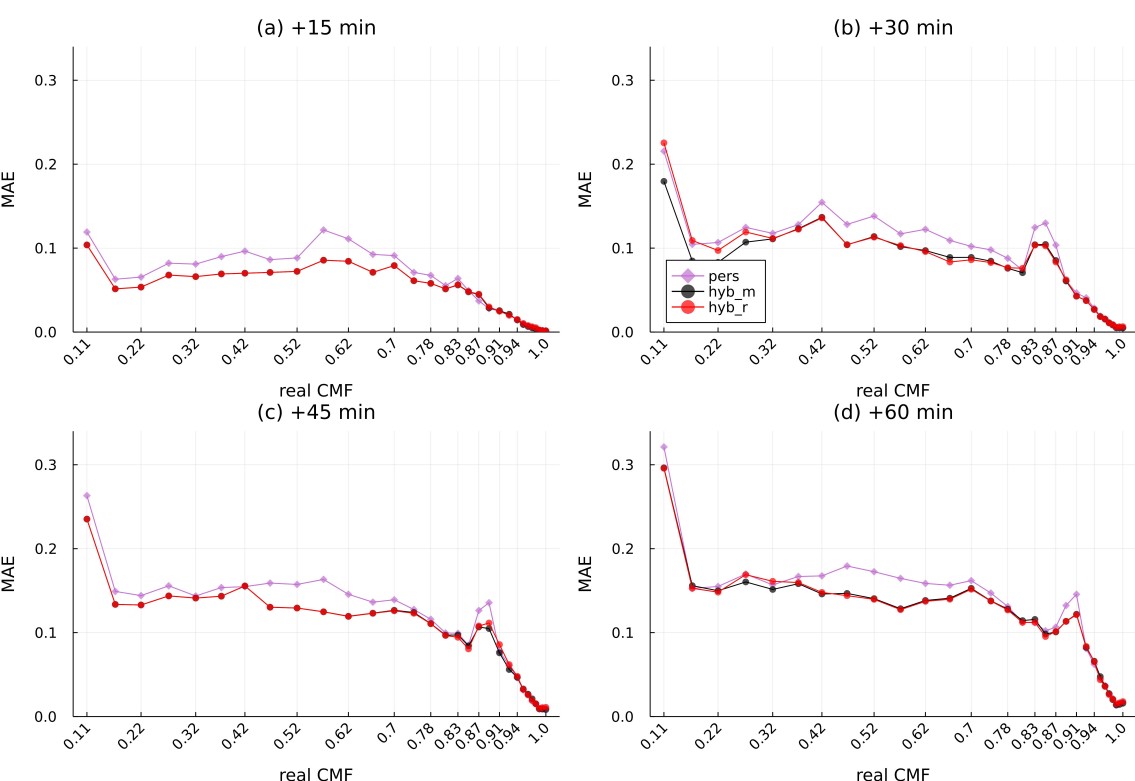

**Figure A5.** MAE for each CMF class for (**a**) 15 min, (**b**) 30 min, (**c**) 45 min, and (**d**) 60 min ahead in 2020 for Bucharest.

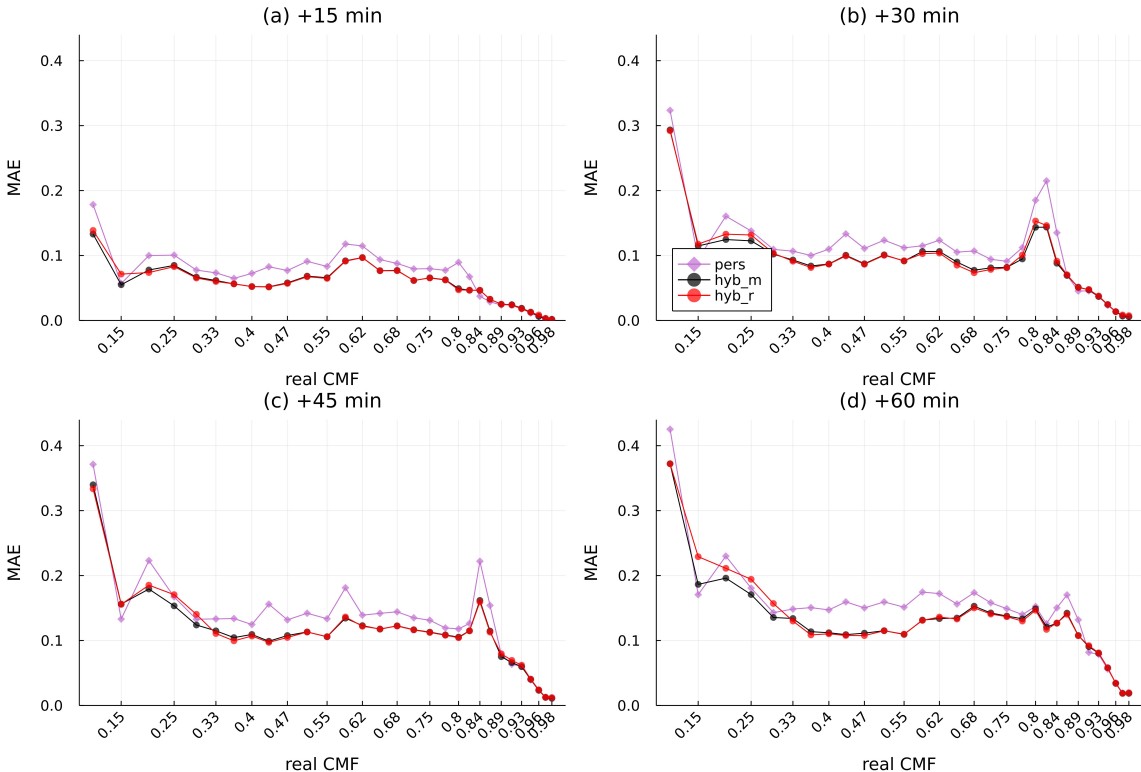

**Figure A6.** MAE for each CMF class for (**a**) 15 min, (**b**) 30 min, (**c**) 45 min, and (**d**) 60 min ahead in 2020 for Helsinki.

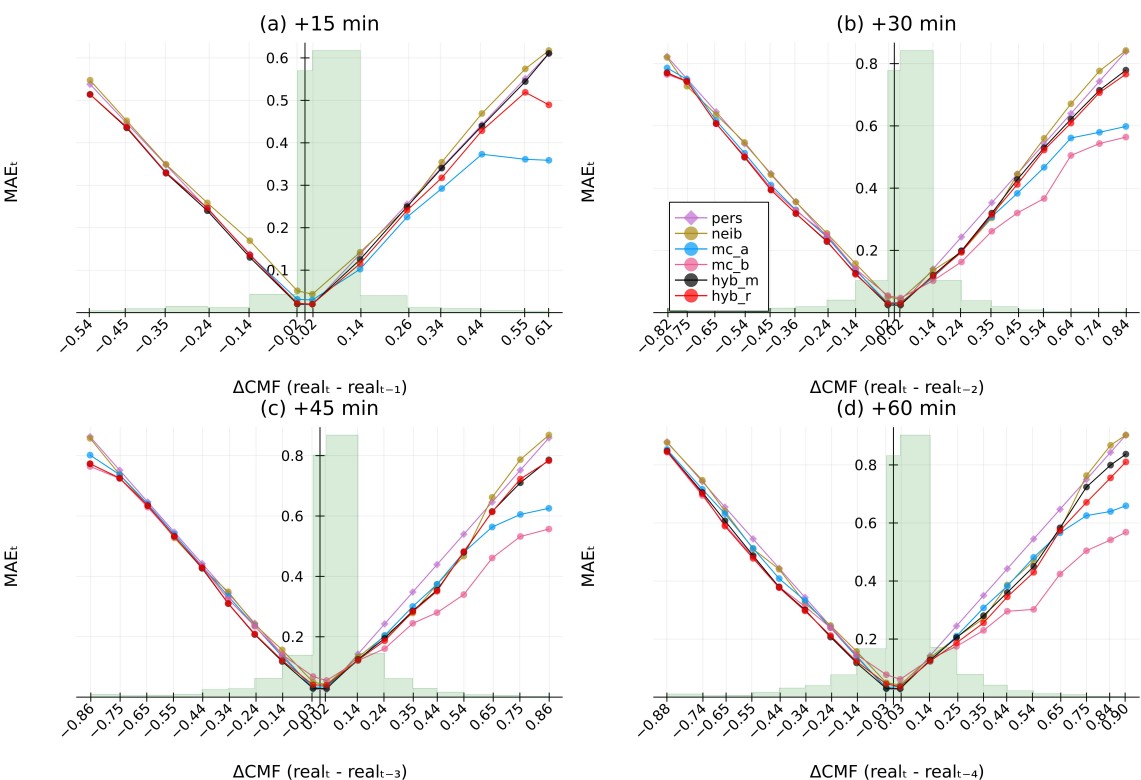

**Figure A7.** MAE of CMF change for (**a**) 15 min, (**b**) 30 min, (**c**) 45 min, and (**d**) 60 min ahead for Athens.

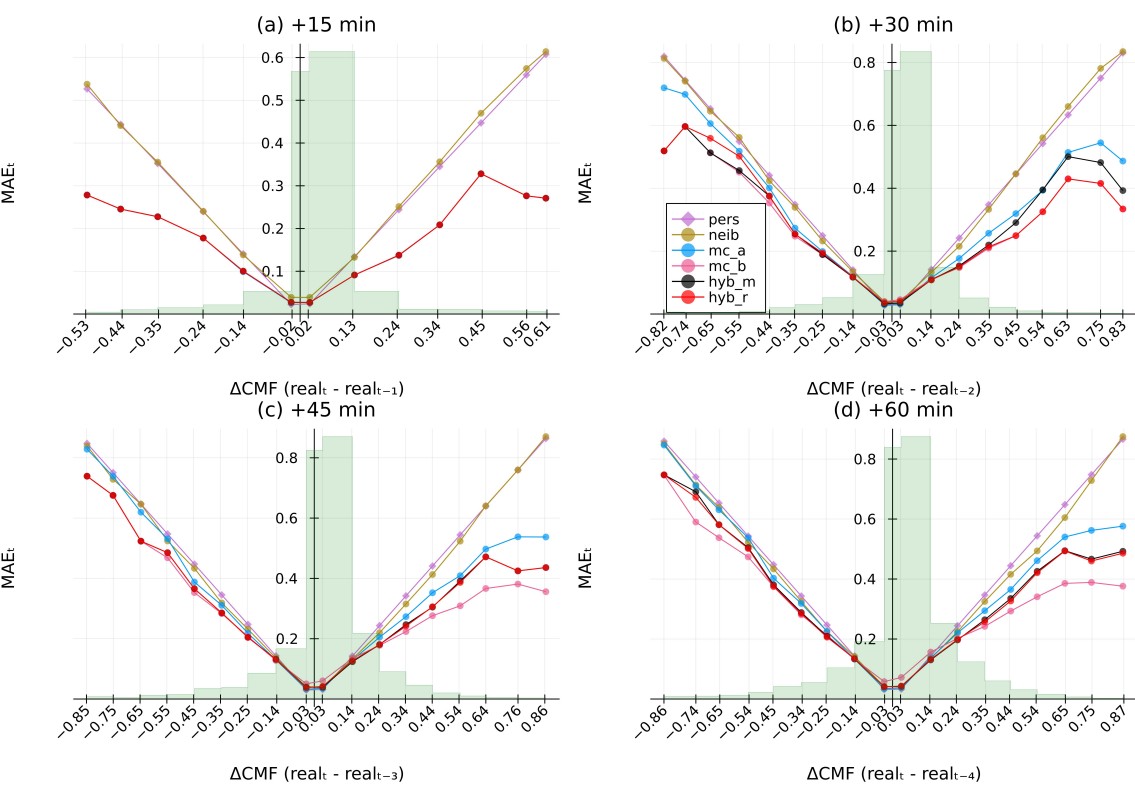

**Figure A8.** MAE of CMF change for (**a**) 15 min, (**b**) 30 min, (**c**) 45 min, and (**d**) 60 min ahead for Bucharest.

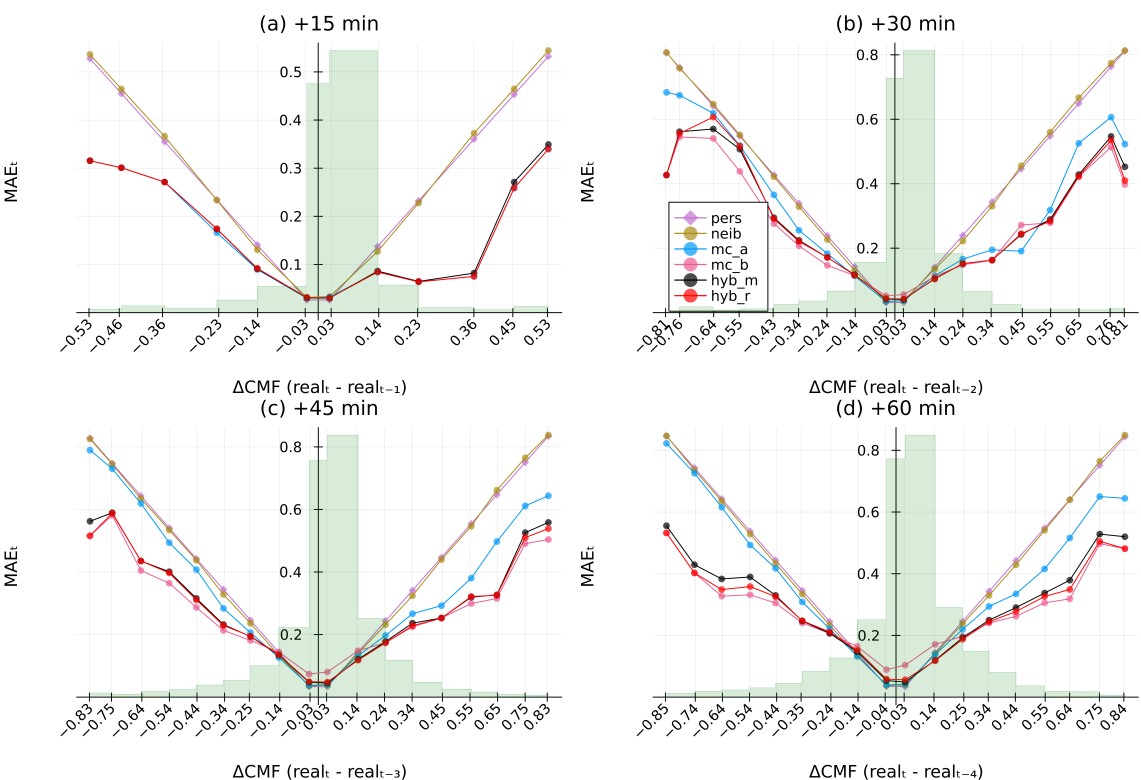

**Figure A9.** MAE of CMF change for (**a**) 15 min, (**b**) 30 min, (**c**) 45 min, and (**d**) 60 min ahead for Helsinki.

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
