# Peer review of "Solar Radiation Nowcasting Using a Markov Chain Multi-Model Approach"

_energies, doi:10.3390/en15092996_

Round 1

Reviewer 1 Report

The authors developed a new solar radiation nowcasting multi-model approach that is simpler and more reliable than the commonly used solar forecasting methods. The proposed model allows solar nowcasting with a lead time of one hour. In addition, the authors' approach of utilizing the irradiance data from the Copernicus Atmospheric Monitoring Service with different clouding conditions improves prediction accuracy. The work is interesting and will be helpful to perform the solar radiation nowcasting if there is no available meteorological data measured on-site. The research question is valid, and the manuscript is clear and well-written. The only weakness I detected was that it could be a little confusing to follow the structure of the global hybrid model developed. Hence, I only have some suggestions:  

  1. In the abstract section is unusual to introduce acronyms. “CAMS” should be removed and mentioned in the introduction section.
  2. “Six other parameters” are mentioned in line 76. They should be listed.
  3. The scales are tiny in Figures 3, 10, 11, 14, 15, A4, A5, A6, A7, A8, and A9. Therefore, they must be enlarged to be able to read it.
  4. Finally, a flux diagram should be included to explain the global hybrid model.

Reviewer 2 Report

Consideration or discussion of effects by altitude on solar radiation prediciton is also suggested. 

Reviewer 3 Report

The paper presents developing a hybrid approach to improve the accuracy of solar nowcasting with a lead time of up to one hour. The method utilized the irradiance data from the Copernicus Atmospheric Monitoring Service and was tested for four European cities with various cloudiness. Several forecast techniques were utilized. The method is presented and discussed in terms of global horizontal irradiation (GHI) prediction for Berlin. It was shown that the new method performed better than the traditional persistence method. Moreover, the new approach has many advantages.

The paper has the correct structure and is well written. The introduction with state of art is appropriate. It described the application of the artificial intelligence methods to predict surface solar radiation and solar radiation time series. The model description is correct and complete. The results are presented clearly and well discussed.

The scope of the paper is suitable for Energies journal. Therefore the paper is recommended for publication with minor revision. The only remark is that figures have too small descriptions, which must be improved.

Reviewer 4 Report

The entire paper needs major revision. It gives the impresion that the author used a computer package on Markov chains and produced some figures which actually are HIGHLY unsuccesfully explained for the reader. The AUTHORS SHOULD MAKE some basic reading on Markov chains of high order.  I SUGGEST Bertchtold and Raftery (2002) Stat. Sci. 17, (3) 328-356 and references within for a start.  By the way they are well known statistical test for the order of a MC (AITKEN). There are certain steps to follow in applying MC. It is not like throwing some data in a black box and whatever comes out in terms of figures is publicable.

I am giving the authors the chance to do a proper job but I am not ready to accept anything less than a good professiomal work.

Round 2

Reviewer 4 Report

There has been improvment in the revision of the paper. However there is still work to be done.  You assume a Homogeneous Markov chain of second order. There are tests for time homogeneity which should be applied see Vassiliou (2010) Chapter 3  John Wiley  Or Anderson and Goodman (1957) Ann. Math. Statist. 28, 89-110; Or Vassiliou (1976) Oper. Res. Quart 27,57-70. Define clearly the states of your MC and provide the transition probability matrices with the appropriate tests. Seasonality or cyclicity I intuitively feel that exist. The least you can do is to provide for the thorough practitioner references useful to tackle the problem.  For example, Gani (1963)  J. R. Stat. Soc. A 126, 400-409. Isaacson and Madsen (1976) Markov chains and Applications. John Wiley. If your model turns up to need a non-homogeneous cyclic Markov  then you should at least refer to Vassiliou (1984) J Appl Prob 21 315-325. Also the citation of Bertchtold and Raftery should be included together with all the above mentioned.  I hope to see all the above done before I supply my final decision.

Round 3

Reviewer 4 Report

The paper has been improved up to a rather satisfactory level, hence I am at the happy posotion to recommend publication. My intuition says that the authors learned useful things for their future work.